# Assessment of Body Condition in a Threatened Dolphin Population in an Anthropized Area in Southeastern Brazil

**DOI:** 10.3390/ani14131887

**Published:** 2024-06-26

**Authors:** Deyverson Silva, Guilherme Maricato, Tomaz Cezimbra, Larissa Melo, Israel S. Maciel, Rodrigo Tardin

**Affiliations:** 1Laboratory of Marine Ecology and Conservation (ECoMAR), Department of Ecology, Universidade Federal do Rio de Janeiro (UFRJ), Rio de Janeiro 21941-902, Brazil; guilherme.713@gmail.com (G.M.); tomaz.cezimbra@gmail.com (T.C.); larissa.vidalmelo@gmail.com (L.M.); oceanmaciel@gmail.com (I.S.M.); rhtardin@gmail.com (R.T.); 2Graduate Program in Ecology and Evolution (PPGEE), Instituto de Biologia Roberto Alcantara Gomes (IBRAG), Universidade do Estado do Rio de Janeiro (UERJ), Rio de Janeiro 20551-030, Brazil; 3Graduate Program in Ecology (PPGE), Instituto de Biologia, Universidade Federal do Rio de Janeiro (UFRJ), Rio de Janeiro 21941-971, Brazil

**Keywords:** anthropogenic impacts, Sepetiba Bay, health, conditioning, marine conservation

## Abstract

**Simple Summary:**

This study investigated the body condition of the Guiana dolphins (*Sotalia guianensis*) of Sepetiba Bay, in southeastern Brazil, between 2017 and 2022. Body condition, a key indicator of animal health, was assessed using standardized photographs taken from a research vessel. These photographs were used to evaluate visible roundness, epaxial musculature, and the prominence of the ribs. This analysis revealed that a majority (68.35%) of the dolphins photographed during the study were in poor condition. This study provides the first assessment of the body condition of the Guiana dolphins of Sepetiba Bay, an ecosystem facing increasing anthropogenic pressures. Our findings highlight the potential impact of human activities on dolphin health, which warrants further investigation.

**Abstract:**

Sepetiba Bay, located in southwestern Rio de Janeiro state, in southeastern Brazil, is a region of extreme anthropogenic impact, and is home to a large population of Guiana dolphins, which face increasing and cumulative pressures on their physical health. Cetacean body condition provides a useful indicator for the evaluation of the conservation status of marine mammals. Given this, the present study quantified the proportion of dolphins with different body condition scores and assessed temporal variation in these scores between 2017 and 2022 through the analysis of photographic records. We analyzed the photographs and identified the individuals using FinFindR and classified each individual based on its apparent body condition. A total of 29,737 photographs were taken during the study, and 79 individuals were identified, of which 68.35% were in poor condition. The evidence suggests that the Guiana dolphins are in relatively poor condition overall, possibly reflecting the cumulative impact of human activities in Sepetiba Bay.

## 1. Introduction

Cetaceans are considered to be environmental sentinels due to their high life expectancy and capacity to indicate disturbances in the ecosystems they inhabit [1]. Many species are top predators with the ability to exert top-down control on biological communities, as well as being keystone species, whose disappearance tends to have a major impact on local food webs [2,3]. Despite their importance for ecosystem health, cetaceans are facing a range of threats as a consequence of the impact of anthropogenic activities on marine systems, worldwide.

Small coastal cetaceans are especially vulnerable to human activities, such as overfishing, shipping, and organic, chemical, and noise pollution, due to their frequent proximity to urban and industrial environments. These activities often act synergistically, leading to cumulative impacts. In these environments, cetaceans are more prone to disease, due to their reduced immunity, which potentially leads to increased mortality, as detected by the number of carcasses washed ashore [4]. Demographic impacts have been observed in a number of populations, including declining abundance and survival rates [5]. External signs of body condition, such as the presence of skin ailments and the prominence of the ribs, may provide vital information about the health, vulnerability, and ecology of cetaceans [6,7]. Monitoring shifts in the body condition of individuals faced with cumulative impacts can provide fundamental insights into the ability of a population to cope with these disturbances [8], and are an important indicator for marine conservation [6,7].

The use of photographs to assess body condition is traditionally carried out on captured or dead animals [9]. More recently, other less invasive methods have been used, such as unmanned aerial vehicles or even photographs taken from a boat [10,11]. The use of photography to assess body condition in dolphins is based on the principle of capturing ideal images of visible morphological features, such as body shape and visible evidence or absence of rib bones. However, studies of small-bodied cetaceans are still scarce [10,12].

For coastal species, such as the Guiana dolphin (*Sotalia guianensis*), the assessment of body condition may provide a valuable tool for population management [10]. Guiana dolphins are small cetaceans that measure up to 230 cm in length, with a maximum weight of 150 kg, no sexual dimorphism and an estimated lifespan of 33 years [13]. These dolphins are found in marine and estuarine waters, including bays, in the eastern tropical Americas, from southern Brazil, in the state of Santa Catarina, to central Honduras [8,13]. In general, this species does not form large aggregations over the long term, but rather at specific events, such as feeding at concentrated resources. Social groups range in size from two to 300 individuals, depending on the circumstances, such as prey availability, habitat quality, and intraspecific competition, which vary considerably among the habitats occupied by this species [10,14,15,16,17].

In Sepetiba Bay, an estuarine environment located in southwestern Rio de Janeiro state, in southeastern Brazil, the local Guiana dolphin population has historically consisted of large groups, with aggregations of up to 300 individuals, often with calves [18,19], which use the area primarily for feeding and breeding [19,20]. The bay is shared by transient individuals (sighted only once) and long-term residents, that have been monitored for more than 10 years [21].

Over the past few decades, this region has undergone intense industrialization and urban growth, resulting in increased exposure to chemical pollutants, domestic sewage, and shipping, and a consequent decline in fish and macroalgal biomass [22,23]. The multiple threats faced by the Guiana dolphins of Sepetiba Bay have resulted in a decrease in group size of more than 50%, a decrease in whistling rates of almost 90%, and more time spent foraging than feeding in recent years [21]. Between November 2017 and February 2018, the population was exposed to the Cetacean Morbillivirus (CeMV), a pathogen that causes immunosuppression, secondary infections, depletion of body condition, and in extreme cases, death. Over a five-month period, CeMV was responsible for the deaths of at least 277 individuals, primarily females and juveniles [24,25,26].

Morbilliviruses are highly contagious pathogens that can cause disease in both humans and animals, with clinical conditions ranging from mild, self-limiting infection to death [27,28]. In cetaceans, CeMV causes damage to the central nervous system, pneumonia, and impacts in particular the maintenance and homeostasis of lymphoid tissues [29]. In Guiana dolphins, the consequences of this infection include an inability to maintain buoyancy, difficulty foraging, disorientation, and susceptibility to secondary infections [29,30]. These problems are exacerbated by poor body condition, which compromises the ability of an individual to cope with the disease [31]. However, little is known about the condition or health of the individuals that were alive in Sepetiba Bay prior to the CeMV outbreak, nor the evolution of the population over the years, given that it was already under considerable pressure prior to the outbreak. 

*Sotalia guianensis* is classified as ‘Vulnerable’ in the Brazilian extinction risk assessment [32], and is considered to be a conservation priority by the International Whaling Commission (IWC) [33] and by the Brazilian National Action Plan for the Conservation of Aquatic Mammals [34]. 

As the population of Guiana dolphins in Sepetiba Bay is one of the most threatened anywhere within the geographic range of the species, the present study aimed to (i) quantify the proportions of dolphins in different body condition classes, and (ii) evaluate the temporal variation in body condition scores during the study period, between 2017 and 2022. If the proportion of body condition scores, an individual indicator, reflects the general health of a population, then we would expect a relatively large proportion of individuals to be in poor condition in Sepetiba Bay, due to the ongoing and chronic impacts that affect this environment.

## 2. Materials and Methods

### 2.1. Study Area

The study was conducted in Sepetiba Bay, a semi-open estuarine environment located in southwestern Rio de Janeiro state, southeastern Brazil. Sepetiba Bay covers an area of 447 km^2^ and is located 60 km to the west of the city of Rio de Janeiro [21,35]. The depth of the bay varies between 2 m and 20 m [36], with salinity of approximately 30 psu, which is lower than the mean salinity of the neighboring ocean, of around 35 psu [37]. The circulation of tides within the bay is influenced by the variation in the directional distribution of winds and associated wind forces [38].

In 2015, a Marine Protected Area (IUCN category V) was created specifically to protect the local population of Guiana dolphins. This protected area also regulates and guarantees the rational use of local natural resources, supports sustainable development practises in the region, and permits activities such as the use of water resources, recreational tourism, fisheries, and research. Sepetiba Bay is home to a rich marine biota and has historically been home to one of the largest populations of Guiana dolphins found within the geographic range of the species [39,40].

### 2.2. Data Collection

The surveys were conducted from a 12-m vessel equipped with an inboard engine between January 2017 and October 2022 as part of a long-term study of *S. guianensis* behavior and demographic patterns. Three fixed routes were surveyed systematically each month in 2017, 2018, and 2019, covering the entire bay (Figure 1). In 2020, the surveys were interrupted due to the COVID-19 pandemic, and the public health social distancing restrictions. Surveys were then conducted in 2021 and 2022 over two systematic routes using a protocol similar to that of the previous years (Figure 1). The routes were changed between 2017–2019 and 2021–2022 due to the inclusion of a soundscape study in the latter period. Despite these adjustments, the modified routes covered the same general survey areas and were designed to maximize the probability of encountering dolphins. Surveys were only conducted in good weather conditions, with wind speeds below Beaufort Sea State 3. The vessel followed pre-established routes to sample the study area equitably at a mean speed of 10–15 km/h. Whenever a dolphin was sighted, the vessel hove to, and the individual was photo-identified by scan sampling at a minimum distance of 50 m [21]. The photo-identification technique is a standard method that is widely used to collect individual information based on the nicks, notches, and other natural marks on the bodies of the dolphins [41]. These markings were photographed using DSLR cameras equipped with lenses of between 75 mm and 300 mm. Two experienced observers then selected the photographs of ‘good’ or ‘excellent’ quality (well-focused, with enough light and minimal blurring) for analysis. This identification process was conducted manually with the support of FinFindR, an R-based algorithm that processes the dissimilarities in dorsal fin contours [42].

### 2.3. Data Analysis

For the analysis of body condition, each individual was assigned to one of three condition classes [10]: good, thin or emaciated. Two or more photographs—up to seven—of each individual were used for this assessment. Only photographs of dolphins completely parallel to the boat (i.e., with their bodies neither emerging from nor submerging into the water) were used for the analysis. This protocol minimized potential biases in the assessment of body condition. For analysis, each photograph needed to capture clearly the dorsal fin and the shape of the body and/or latero-dorsal concavity, for the reliable assessment of body condition. The body condition of each individual was scored using the following criteria [based on 1,2]: (1) good condition—rounded body shape, (2) thin—reduced epaxial musculature visible through the latero-dorsal concavity, and (3) emaciated—prominent ribs and thin blubber (Figure 2). This analysis included only photographs in which the body shape and/or latero-dorsal concavity were clearly visible, for the reliable assessment of body condition.

The temporal variation in body condition was assessed for individuals that were recaptured (photographed) on at least two different days during the study period, preferentially, in different years. To verify the effects of the CeMV outbreak, the study period was divided into three sub-periods, representing the moments ‘before’ (January through October 2017), ‘during’ (November 2017 to March 2018), and ‘after’ the outbreak (April 2018 to October 2022) [43].

## 3. Results

Sampling was conducted on a total of 61 days in Sepetiba Bay, which permitted the collection of a comprehensive set of photographic data over surveys lasting a total of 366 h. A total of 226 photographs were used for the analysis of the body condition of the Guiana dolphins, allowing for the identification of 79 individuals.

More than one-third (35.44%) of these individuals were classified as “thin” (BCS2), and 32.91% were evaluated as “emaciated” (BCS3). Only 26 individuals (32.91%) were in good condition (BCS1). Three individuals were recaptured in different years, permitting the longitudinal assessment of their body condition. The individual SEP757 was first photo-identified on 2 March 2018 in a ‘thin’ condition and was recaptured in the same condition on 16 October 2019, while SEP637 was first photo-identified on 17 March 2017, in an ‘emaciated’ condition and was recaptured in the same condition on 16 October 2019, and SEP805 was first photo-identified on 24 October 2018 in ‘good’ condition and was recaptured in ‘emaciated’ condition on 16 October 2019.

A total of 61 individuals were assessed in the rainy season (austral spring and summer) and 18 in the dry season (autumn and winter). The frequency of unhealthy individuals (“thin” or “emaciated”) was 67.21% in the rainy season and 61.11% in the dry season. 

Comparing the periods before, during, and after the CeMV outbreak, a third of the dolphins were in good condition both before and after the epizootic period, but none were identified in good condition during the event (Table 1). After the outbreak, more than half of the dolphins were classified as thin (Table 1).

## 4. Discussion

Most of the Guiana dolphins photographed prior to the morbillivirus outbreak were in poor (thin or emaciated) condition, a trend that continued after the outbreak. The large proportion of individuals photo-identified in a poor condition is a preoccupying reflection of the health status of this dolphin population [3]. The poor condition of the dolphins is likely a consequence of the cumulative impacts on these animals over the past decade in Sepetiba Bay, resulting from overfishing, excessive tourism, and noise, chemical, and organic pollution [17,37]. It is important to note, however, that this dataset includes transient individuals, which may be less exposed to these impacts than the resident dolphins.

Over the past three decades, the fish community of Sepetiba Bay has undergone significant changes, in particular the species of the family Sciaenidae, which are an important group of prey for the Guiana dolphins. The most pronounced alterations occurred in the inner portion of the bay, which has the most intensive anthropogenic impacts [22,23]. Overfishing by industrial vessels and the high concentrations of pollutants, such as chromium (Cr), cadmium (Cd), manganese (Mn), lead (Pb), and copper (Cu) [44] may have had negative effects on many of the fish species that are important components of the diet of Guiana dolphins, in particular, whitemouth croaker (*Micropogonias furnieri*), acoupa weakfish (*Cynoscion acoupa*), banded croaker (*Paralonchurus brasiliensis*), Lebranche mullet (*Mugil liza*), and Atlantic anchoveta (*Cetengraulis edentulus*) [44,45,46]. The decreased availability of this prey [47] most likely affected the body condition of the dolphins, which would have threatened the population when exposed to the disease. 

Thin and emaciated individuals may be affected by shifts in their physiological or behavioral capabilities. Poor body condition resulting from reduced adipose tissue will affect the thermoregulatory capacity of the dolphins, and reduce their access to energy, given that stored lipids are oxidized for energy production [48]. Thin or emaciated individuals are also more likely to have reduced immunity because of the greater demands on their energy stocks, which may result in greater metabolic effort, potentially leading to a reduction in foraging efficiency [49] and an increased susceptibility to secondary infections, due to metabolic stress [50].

Like other delphinids, Guiana dolphins have late sexual maturity (females mature between five and eight years of age and males at around seven years old), and a gestation of 11–12 months [51]. The combination of this slow reproductive cycle with insufficient supplies of adipose tissue raises major concerns for the females, in particular, and for the development of their offspring. Poor body condition in females can lead to (i) low ovulation rates, which reduce the chance of conception [52]; (ii) a longer inter-birth interval, reducing the population’s reproductive potential [53], and (iii) a reduction in the quantity and quality of the mothers’ milk due to their low energy reserves, leading to a potential reduction in offspring viability and survival [54]. Given the critical role of Sepetiba Bay as a breeding zone and nursery ground for the dolphins, the local anthropogenic impacts pose a significant threat for the population’s health and potential for long-term survival. Over the long term, in fact, a population with a high proportion of individuals in poor condition may experience decreasing survival and reproductive success, with significant impacts on population structure. These tendencies represent increasing challenges for the protection of the Guiana dolphins of Sepetiba Bay. However, long-term monitoring must be conducted to substantiate these conclusions.

The presence of commercial shipping, and fishing and tourist boats in areas where food is available can have a negative impact on the body condition of Guiana dolphins, increasing the risks for their health. Foraging in areas occupied by vessels may mask dolphin communications, which are essential for the coordination of the group’s movements when catching prey [55]. A previous study in Sepetiba Bay [17] found that Guiana dolphins communicated significantly less in noisier areas. Dolphins in poor condition may prefer to conserve energy by avoiding communication rather than spending more energy to increase vocalization emission rates, even if this may affect their feeding success [43]. Entanglement in fishing nets may exhaust emaciated dolphins rapidly due to their low energy reserves, leading to reduced thermoregulation, rapid muscle fatigue, and a decline in swimming capacity, which can result in drowning [56].

In Paranaguá and Laranjeiras bays, in the southern Brazil state of Paraná, the body condition of the local Guiana dolphins appears to be impacted by a range of human activities, especially in the case of females with calves. Like Sepetiba Bay, Paranaguá Bay is subject to intensive and cumulative human impacts, such as shipping, overfishing, and tourism [10]. In this population, the dolphins, with or without calves, presented a range of pathologies that were associated with poor body condition, and the authors concluded that calves in poor condition were subject to inadequate nutrition, which led to low immunity and increased vulnerability to dermal pathologies, given the local presence of potentially pathogenic microorganisms.

During the CeMV outbreak in Sepetiba Bay, more female and immature Guiana dolphins died than adult males [57]. In mammals, body condition has an important protective role and complements metabolic activity [58]. Individuals in poor condition may perform poorly in vital activities, such as foraging and parental care, which require accentuated lipid metabolism [57,59]. More than 60% of Guiana dolphins in Sepetiba Bay were reported to have been in poor condition in the period prior to the outbreak [43]. This indicates that the dolphins were likely already immunodeficient, as a result of the cumulative human impacts in the bay. During the outbreak, none of the dolphins observed were in good condition, with 75% being thin, and 25%, emaciated.

There was a simultaneous outbreak of CeMV in 2017–2018 in Ilha Grande Bay, which is adjacent to Sepetiba Bay. Ilha Grande is a more pristine environment, with fewer anthropogenic impacts in comparison with Sepetiba Bay. While the body condition of the individuals from Ilha Grande Bay was not assessed, approximately four times fewer carcasses were found during the morbillivirus outbreak in comparison with Sepetiba Bay [60]. This reinforces the conclusion that the intensive human impacts in Sepetiba Bay are a significant driver of the probable immunosuppression of the local dolphins.

New outbreaks of disease and mortality related to the poor health of the Guiana dolphins, driven by the intense impacts in Sepetiba Bay may recur, given that human activities continue to intensify. In this context, it will be essential to continue the systematic monitoring of the local Guiana dolphins over the long term, and to devise effective measures to mitigate human activities in Sepetiba Bay. Reinforcing the health of the Guiana dolphins would increase their overall resilience and help to prevent further impacts on the health status of the local population [10].

Despite being a traditional and non-invasive method, the photographic analysis of body condition has certain limitations: (i) lighting conditions during fieldwork may alter the perception of depth on the dolphins’ bodies, (ii) differences in the experience and fieldwork capabilities of the researchers may influence the number and quality of the photographs obtained during surveys, and (iii) the variation in the position of the animals when being photographed could potentially lead to misinterpretations of body condition. Even so, efforts were taken in the present study to minimize these potential biases by (i) using only good- or excellent quality photographs with sufficient lighting and focus to ensure accurate assessment, (ii) using photographs obtained by the researchers most experienced in photo-identification, and (iii) selecting photographs in which the dolphins were completely parallel to the boat. Acknowledging and resolving these limitations will be crucial for the enhancement of procedures for future studies, ensuring a more reliable analysis of body condition, not only for Guiana dolphins, but also for other small and medium-sized cetacean species.

It is also important to note that, due to logistic limitations and financial constraints, the sampling effort was not distributed regularly among the study years. It was possible to conduct more consistent surveys throughout the year prior to the morbillivirus outbreak. Sampling effort was reduced in 2018 and 2019, due to problems with the survey vessel and the need for the research team to assist with the monitoring of beached carcasses resulting from the morbillivirus outbreak. The COVID-19 pandemic also affecting sampling in 2020. Clearly, this heterogeneity in sampling effort may have introduced a certain amount of bias, although, as one of the principal objectives of the present study was to determine whether the dolphins were already in poor condition prior to the outbreak, the total number of samples (*n* = 79) can be considered to be representative of the population as a whole. The number of recaptured individuals was extremely small (*n* = 3), however, although this is an intrinsic characteristic of the population, where transient individuals coexist with long-term residents [13]. 

This study presents the first systematic assessment of the body condition of the Guiana dolphins of Sepetiba Bay, a type of analysis limited by three principal factors: (i) Guiana dolphins, unlike other coastal dolphins, are small, have a cryptic behavior, and do not typically have large, conspicuous marks on their dorsal fins [61]; (ii) the rigorous selection of photographs with the dorsal fin (for individual identification) and body contours in the same frame, which limited sample size, and (iii) the residency of the Guiana dolphins in Sepetiba Bay is naturally low, leading to typically low recapture rates [18]. Even so, the study provides the first assessment of individual health in a population facing increasing anthropogenic impacts, which appear to have caused significant behavioral changes, such as decreased group size, reduced whistle emission rates, and declining feeding patterns [21].

## 5. Conclusions

The results of the present study highlight the large proportion of individuals in poor body condition in a population affected by the accumulation of multiple anthropogenic impacts. Deciphering the dynamics of the variation in body condition in the context of habitat shifts should support the development of effective measures that contribute to the improvement of the health of these dolphins. Understanding the interplay between anthropogenic impacts and the health of the Guiana dolphins will also be crucial for the proactive intervention of decision-makers. The present study provides important insights into the health of the Guiana dolphins that may encourage stakeholders to mitigate the impact of their activities in Sepetiba Bay, which will, in turn, help to protect the local Guiana dolphin population.

## Figures and Tables

**Figure 1 animals-14-01887-f001:**
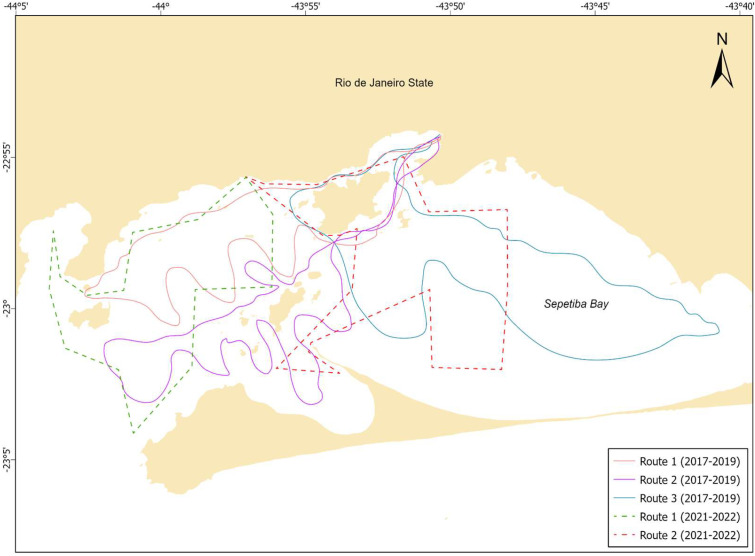
Map of the routes surveyed in Sepetiba Bay between 2017 and 2022.

**Figure 2 animals-14-01887-f002:**
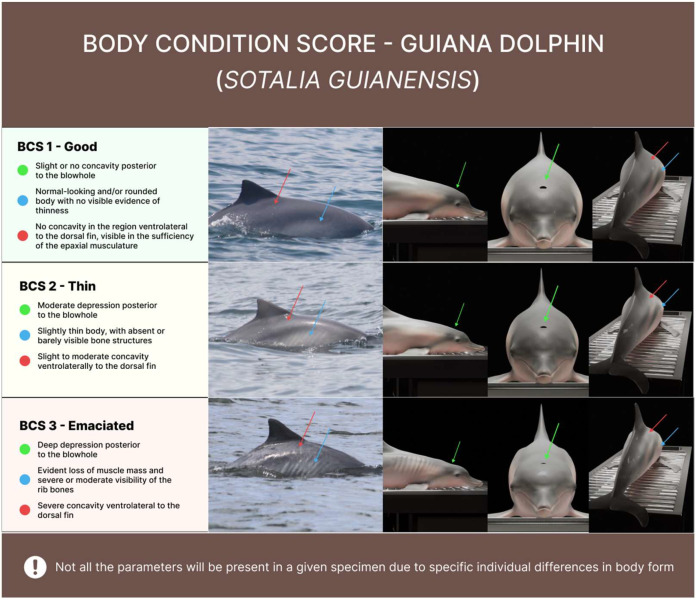
Representative photographs and 3D model of the Body Condition Scores (BCSs) used to assess body condition in the Guiana dolphin population of Sepetiba Bay [10]: BCS1—Good, normal, rounded body shape; BCS2—Thin, with a distinct latero-dorsal concavity, and BCS3—Emaciated, with clearly-visible ribs. The images illustrate the three scores used to classify the body condition of the animals, and can serve as a guideline for the classification of the BCS of other cetacean species. The high-definition BCS image is included in the Appendix A.

**Table 1 animals-14-01887-t001:** Proportion of Guiana dolphins identified in the different body condition classes before, during, and after the cetacean morbillivirus outbreak in Sepetiba Bay, southeastern Brazil. The assessed individuals were not necessarily recaptured in different periods.

Body Condition	Number (%) of Dolphins in the Period:			Total
	Before	During	After	
good	9 (37.50%)	0	6 (33.33%)	15
thin	6 (25.00%)	6 (75%)	10 (55.56%)	24
emaciated	9 (37.50%)	2 (25%)	2 (11.11%)	13
total (*n*)	24	8	18	52

## Data Availability

The original contributions presented in this study are included in the Appendix A.

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
