# Peer review of "Assessment of Body Condition in a Threatened Dolphin Population in an Anthropized Area in Southeastern Brazil"

_animals, 2024, doi:10.3390/ani14131887_

Round 1

Reviewer 1 Report

Comments and Suggestions for Authors

Review of: Body condition assessment of an endangered dolphin population in an anthropized area in southeastern Brazil

This manuscript describes a study of Guiana dolphin body condition based on photographs analysis. Despite the numerous issues, I enjoyed reading this manuscript. This study represents a very useful contribution to the field, we need ways to evaluate body condition in dolphins for conservation purposes. However, the problem is that no reliable way to assess dolphin body condition on live animals has been developed yet. Until now, only data from UAVs have been proven reliable since ratios can be calculated. Using photographs is much more convenient but is associated with many biases, which is why no/few study successfully used it on wild dolphins yet. Very poor body conditions may be obvious on pictures, but the difference between good body condition and thin body condition is much harder to find. In addition, what is called « good » body condition may actually not be the optimal body condition, but only an acceptable condition. Without data collected on stranded animals, captive animals, or captured animals, it is impossible to affirm that this « good » body condition is actually optimal. In my opinion, we can only say that it is better than the « thin » and « emaciated » conditions.

In addition to these biases, the discussion of the results has to be entirely worked on. Authors over interpret results and provide information and arguments that are useless or even spurious.

All that said, I believe that we still need this kind of initiatives, even though biases are present. In order to publish this study, you need to acknowledge all the biases and be very conservative in the way you label and interpret the body condition categories. You can have a look at other papers on dolphin body condition to check the labels they used (e.g., Serres et Al., 2024). You also need to work on your discussion to focus on your results and present your hypotheses regarding the causes for the observed bc and its consequences clearly. This requires a lot of work, but I am confident that authors can work on this and make the manuscript acceptable for publication. I provided detailed comments to help authors understand my points and modify their manuscript.

Detailed comments:

The English level is good, but some grammar/vocabulary mistakes should be corrected. I listed some below but did not correct all.

L12: These photographs were used to evaluate

L14: photographed dolphins

L19: subject to high anthropogenic impact

L21: I am not sure « strong » is the most suitable adjective here. Maybe « useful », « informative » or something like that?

L25: I am not sure the number of pictures is relevant. Within these pictures, some were probably not useful (e.g. tail pictures, bad angles, blur). And one ID may have 10 pictures while another one has 1. Also, a very good photographer may just need one or two pictures to capture an animal well enough while a less experienced photographer may need many more. Just to say that I don´t think this information brings anything to the paper.

L28: You did not study behaviour here, right? If not, please remove the mention of it.

Introduction:

The context and species are well introduced but I suggest spending less time on general information globally and providing more information about body condition assessment in cetaceans. There are plenty of studies conducted using UAVs, most on larger whales, but also some on dolphins that are experiencing similar conditions than yours (e.g. Sousa chinensis). Currently the introduction contains a lot of information but it is not perfectly clear what the goals are and how body condition assessment will help you answer your questions. Removing some useless info and adding info that is really related to your study will help.

L33-41: No mention of fishing activities? These activities are potentially also linked with cetacean health in many ways, including disturbance, reduction of fish availability etc.

L70-77: I think we do not need all this species information here since it is not related to your study.

L90: more time spend foraging than feeding? How do you differentiate? Can you actually see the dolphins catching the fish versus searching for it?

Material and Methods:

L121-131: I believe this information belongs to the introduction.

L145-147: This belongs to the data processing part. Here you should just state that onboard photographers took pictures of dolphins.

L155: into three categories

L155-161: It is hard for me to figure the difference between good and thin conditons on photographs… Especially since the angle of the photographs and posture of the dolphin may differ greatly among photographs.

Also, it is not mentioned if only one photograph was used per ID sighting or if all available photographs were used. From my experience, an ID may look more thin on a picture than on another taken during the same encounter…

Please provide more information about the photographs analysis and examples of photographs categorized as good, thin, and emaciated (not just one picture, we need more to see the difference)

Figure S2: nice figure but it does not help us understand how you categorized dolphins using photographs. We need a figure to show us examples of photographs for each category.

Also, what was this figure based on? The work from Joblon et Al.? Or do you have data from stranded/captive animals? Or was the figure made using the photographs?

Results:

L179: That is what I said above, only the number of pictures used for the study is informative.

However, since you stated you got 134 IDs and only 280 photos, it looks like you used only one single photo for assessment in each sighting and each ID was sighted around 2-3 times during the study period. While the relighting rate is ok, using a single photograph for body condition assessment really makes me doubt the results. How were these photographs selected? Based on what criteria? We need at least more information on that point.

L181: in the methods you say that only individuals that were sighted at least twice were included. Here you state that only 7 individuals were recaptured.

Also, be careful, one single association with a calf does not mean the adult is the mother. Calves are often observed swimming with dolphin females other than their mother. Usually we only assume that the adult is the mother when we have seen two or more close associations during different sightings which I believe you don’t have here.

L184-185: you are already starting to interpret your results. This is not the place for it. And that is also why I suggest that you re-label your categories to remain conservative.

L194: where is the figure 4?

L195: one ID went from good to emaciated in 28 days? That is hard to believe and confirms my doubts about the categorization. The virus may have contributed and I am not a vet though, so these are only thoughts. If you have vets confirming that this may happen that fast, please forget my words.

L196: you did not define seasons in the material and methods, please add that.

L204-207: this paragraph could be written in a more straightforward way.

Table 1: it would be useful to have this kind of information for the season and the females with calves too. In the text you only provide percentages, the table allows to see raw numbers too.

In the table, it is obvious that the number of IDs is much lower during and after than before the epidemic. This balance issue has to be discussed.

Discussion:

The discussion is way too long because results are often over interpreted and information that is not directly linked to it is often provided. It can be shortened significantly by removing the useless information, focusing on the results, and better organizing. There is a clear lack of link made between information available in the literature and the results of the study. I suggest splitting the discussion into two parts: (1) why is the body condition that low and (2) what are the potential consequences of such a low body condition.

L215-221: the link between body condition and an infection is not clear here. It starts with a statement on body condition and continues with two sentences about immunity and infection without any link…

L222-230: No link with the study is made here, please use the provided information in the context of your study.

L232-241: Here we have some link with the study but it is unclear what is the cause of what. Is your hypothesis “the virus is responsible for the poor body condition”? Or “the low fish abundance is responsible for the poor body condition”? Or “the contamination through the food chain is responsible for the poor body condition”? Or all that?

L242-248: Same than above; no link between the information provided (PCB concentration) and the actual results (poor body condition).

L250-256: I do not think this is the best explanation for the absence of seasonal pattern in body condition… the temperature usually plays a role, as well as the prey composition in terms of nutrients.

L271-277: Similar than above, I can see the point but it is far to be clear.

L379-284: I feel like you are trying to include all the factors that may affect dolphins in the discussion but not all are directly linked to your study. You are going too far here, all animals can get injured and hit by boats or entangled. Do you have a reference stating that risks are higher for thin animals? If not, I suggest not going into that discussion.

L294-299: the percentage of potentially lactating females with poor bc is not higher than for other individuals, therefore your arguments make no sense here. You try to explain something that is not a result…

L301-304: it increases competition only if low bc is caused by a lack of resources and you don’t know if it is the case here.

L305-311: Same than above; statements without links with each other and with the results.

L318-324: what is the link with the study?

L334-335: this aspect is very important for the interpretation of the results and should be discussed more.

L348-349: That is not true. This study reveals poor body condition in a population that is subject to human activities. However, you do not have data to prove the link between human activities and poor bc.

Comments on the Quality of English Language

The english language requires moderate improvement.

Author Response

We would like to thank you for the excellent review and detailed comments you provided on our work. Your observations were crucial in improving the quality and clarity of our manuscript. More details can be found in the attached document.

Reviewer 2 Report

Comments and Suggestions for Authors

General comments

This manuscript described the body condition of Guiana dolphins in Sepetiba Bay and was a pleasure to read, easy to understand and although some further work is required, this manuscript will be a great contribution to literature. Additionally, the results of this study can be used to inform conservation management of this species and function as a baseline to which future studies can be compared to. 

·     1. I think there should be some testing as to the camera angle and the animals’ posture and how these affect the perceived body condition in the resultant images. In my experience, animals who are healthy and in good body condition can look thinner in images where they are for example stretching or mounting another dolphin. I think you should have a standard image angle and proportion of the body visible from which the body condition could be reliably evaluated. Using a standard like this would ensure that at least within your study your results for each individual are comparable. Images in your supplementary material should be part of the main manuscript in my opinion. It is good to see how clearly you can identify an emaciated individual. Your supplementary material could include a variety of images that were classed as ‘good’, ‘thin’ or ‘emaciated’.

·       2. To me, your results indicate that the population in general had more individuals with lower body condition score during the CeMV than it did before and after, where the proportions between good vs poor body condition individuals were similar. This suggests that the CeMV is responsible for lower body condition than are all the accumulative impacts. Of course, the fact that such large proportion of individuals would generally have poor body condition, makes it concerning, and perhaps this can be attributed to cumulative impacts in the Bay specifically, but then again maybe not if the animals are transient as indicated in the discussion. The methods are not water tight (see my specific comments below), which makes it hard to evaluate the robustness of this study. I think more stringent photographic scoring protocol is required and perhaps less emphasis to be put on the CeMV but on the generally poor body condition observed in this population, and that this may in fact make them more vulnerable to CeMV when it cycles back.

·       3. The discussion is very speculative. Be careful not to go beyond what your data shows and be very clear that these are possible contributors to poor body condition and make sure you specify how each threat translates to poor body condition. Some more information on general mortality rates would be great addition, and whether observed poor body condition often results in mortality.

·       4. In your methods also provide a standard image you would use for analysing body condition.

Introduction

Line 32-41: This first paragraph of the introduction is quite confusing given the manuscript is body condition of dolphins. Actually all the text up until line 52 seems out of place. Your introduction would best start with line 53.

Line 56-58: This is not a well linked sentence. How does body conditions provide this vital information. I think it would be best to elaborate here given the main subject matter of the manuscript.

Line 70: I think you could lose the start of the sentence before the colon.

Line 110-112: This confuses me. Is the aim of the study to see whether CeMV infection is implied by poor body condition?

Materials and Methods

Given photographs were used to study body condition, I would have thought that the methods of photographing individuals would have been specific to meet this aim. For example, photographs of dolphin postures in Figure 2. Is this study one that was thought to be conducted after the data had already been collected? If so, it is best to state this.

2.2. Data sampling: Why was the study design changed after the pandemic? Not that the study design is necessarily pivotal to this particular study, it would be great to know how the study routes were designed and how designs were run to ensure equal capture probabilities.

Line 143-145: Positional information is not necessary as it’s not relevant to your study.

Line 146: I assume you are meaning widely used? The word ‘used’ is missing.

Figure S2: Should this not just be Figure 2?

Line 163-164: Was there are a criteria how temporally separated the two sightings would have to be?

Line 167-168: I don’t think that this is a sufficient determination for individual being female. Perhaps better would be an adult consistently observed with a calf in baby position was assumed a female and if the calf association persisted over multiple sightings, a confirmed female.

Results:

Table 1. The fact that you have observed a lot less animals during the CeMV is problematic. Also you have in total captured about a half of the population for body condition evaluation. Is there a possibility that the sampling affected these results?

Discussion

Line 333: But Table 1 states only 69 individuals? I am a bit confused what Table 1 is for as you do mention earlier you had 134 individuals who were the ones included in this study.

Line 335: If you included transient individuals in the study, then the cumulative impacts on the dolphins in Sepetiba Bay would not necessarily be valid?

Comments on the Quality of English Language

Generally, the English language was grammatically correct and easy to read. Some odd phrases were used here and there. The writing could be very much more concise and structured better for flow.

Author Response

(The authors gave the same response as above.)

Round 2

Reviewer 1 Report

Comments and Suggestions for Authors

2nd Review of  “Body condition assessment of an endangered dolphin population in an anthropized area in southeastern Brazil”

I have now read the new version of the manuscript and I would like to thank authors for taking the suggestions I made into account and modifying their article.

The manuscript has improved but is still far to be ready for publication. The English grammar needs extensive revision. If the journal offers an English language editing service, please take it; if not, please find a native English speaker to correct your manuscript. In addition, there are still important issues in organization, flow, and presentation of information more globally. What I mean is that most of the required information is here but presented in a wrong way (wrong order, no link with the data made, etc.). I do not have time to engage again in another very detailed review like the first one, and I believe this is the role of the senior authors of this paper to do such work before submission. Therefore, I encourage the senior authors of this manuscript to carefully check it if possible and help the younger ones to present their study better. I still believe this study is an important contribution to the field, this is why I recommend Major Revisions again.

Here are very few comments:

L21: Replace “of marine ecosystems” by “of marine mammals”

L23: Replace “We analyzed the photos using FinFindR software to identify dorsal fin marks, and then manually defined recaptured individuals and classified their body condition” by “We analyzed the photos and identified individuals using FinFindR and categorized them depending on their apparent body condition”

L24: “A total of” ??? Something is missing here

L25: Remove “of the population analyzed”

L94: “In addition, this problem is exacerbated by poor body condition, which compromises the individual’s ability to recover and cope with a CeMV epidemic.” Do you have any reference for this statement?

L135: “a scan followed”?

Figure 2: I am sorry, but the pictures you added are completely contradicting what you say in the text… Yes, dolphins are parallel to the boat, but their body posture is completely different… The first and last one have the front of the body parallel to the surface and the tail under while the second one is completely arched. How can you compare these pictures when the dolphin is in a different position? It is a huge bias. The first and second dolphins could have the same body condition.

The pictures in the Supplementary material are much better even though there are still differences in body posture that may influence the categorization. Please use these pictures in the manuscript!

L176: Still mentioning the total number of photos… No use.

L178-180: For the recaptured ones, which sighting did you use to calculate these percentages? Maybe better to present percentages for “sightings” instead than for “individuals”.

Table 1 : Same comment than above, is it for “sightings” or “individuals”? Which means did you include only one or all sightings for the recaptured IDs?

L271-277: No link with your study. You did not study immunology parameters.

L285-294: Tell us the biases of YOUR study, what did you do to prevent it, what biases are still present, do they affect your results, etc.

Comments on the Quality of English Language

The English grammar needs extensive revision.

Author Response

Thank you for your comments on this second round. Detailed comments are attached. 

Reviewer 2 Report

Comments and Suggestions for Authors

Dear authors,

Your manuscript was much improved and I commend you for adding more clarity in your methods especially. I still feel some minor edits should be made - please see the attached file with my comments/suggestions in red. 

Author Response

(The authors gave the same response as above.)

Round 3

Reviewer 1 Report

Comments and Suggestions for Authors

I would like to thank the authors for the revision they provided, the manuscript was significantly improved.

Comments on the Quality of English Language

The English language was improved a lot. Even though some mistakes may still be present, I do not have the ability to correct the language much further.